# Long Work Hours, Overtime, and Worker Health Impairment: A Cross-Sectional Study among Stone, Sand, and Gravel Mine Workers

**DOI:** 10.3390/ijerph19137740

**Published:** 2022-06-24

**Authors:** Aurora B. Le, Abdulrazak O. Balogun, Todd D. Smith

**Affiliations:** 1Department of Environmental Health Sciences, School of Public Health, University of Michigan, Ann Arbor, MI 48109, USA; 2Safety and Occupational Health Applied Sciences, Keene State College, Keene, NH 03435, USA; abdulrazak.balogun@keene.edu; 3Department of Applied Health Science, School of Public Health, Indiana University Bloomington, Bloomington, IN 47405, USA; smithtod@indiana.edu

**Keywords:** burnout, health impairment, overtime, stone, sand, and gravel miners, work hours

## Abstract

Background: Research has shown that long work hours and overtime are associated with health impairment, including stress, burnout, and overall health. However, this has not been thoroughly assessed among stone, sand, and gravel mine workers. As such, this study examined whether significant differences in stress, burnout, and overall health existed among workers that worked different hours each week. Methods: ANOVA analyses were completed for the outcome variables (stress, burnout, and health status). Each analysis included three categorical independent variables: age, sex, and work hours. Age and sex were control variables. BMI was added to the health status analysis as an additional control variable. Results: There were significant differences between work hour groups for all three outcomes. Post hoc analyses determined that workers working >60 h/week had more stress, more burnout, and lower health. Differences were not found between age or sex. There were no differences in health status for different BMI groups, but the interaction of BMI and work hours was significant. Conclusions: Working more than 60 h per week was problematic. Mine and safety administrators should enact programs to protect and promote worker health, particularly among those working long hours, especially if more than 60 h per week.

## 1. Introduction

Long work hours and overtime have been linked to safety issues and health impairment. The existing literature supports the positive relationship between long work hours, overtime, and worker health impairment, i.e., physical and mental health symptoms [1,2,3,4,5,6,7]. Work hours and overtime are components of job demands. According to Bakker and Demerouti’s Job-Demands Resources (JD-R) model, job demands are physical and emotional stressors, e.g., time pressure, heavy workload, stressful working environment, poor relationships. Job resources are physical, social, or organizational factors that help workers achieve goals and reduce stress, e.g., autonomy, mentoring, strong work relationships [8]. When job demands are high and job resources are low, this results in health impairment such as poor health outcomes, stress, and burnout [8,9].

Workers in stone, sand, and gravel mining (SSGM) operations are an under-researched population related to long work hours and overtime, despite SSGM being the largest mining sector in the United States. The SSGM sector accounts for approximately 80% of all mining operations, with more than 100,000 workers working at more than 10,000 U.S. sites [10]. Furthermore, SSGM workers often work 10–12 h per shift performing jobs common to the industry, including material handling, equipment operation, driving, maintenance operations, or administrative work [11].

Long work hours and overtime have been associated with poor health outcomes, such as poor sleep quality and deprivation, neurocognitive decline, low-grade inflammatory responses, cardiovascular and immunologic decline, smoking, increased errors, greater incidence of injuries, and increased sick absences [4,12,13,14,15,16,17]. Musculoskeletal symptoms (MSS) and musculoskeletal disorders (MSDs) are notable health impairment issues related to long work hours and overtime. Studies show that long work hours and overtime result in greater MSS and MSDs among healthcare workers [12,18]. MSS and MSDs particularly affect worker populations, like SSGM workers, who are chronically exposed to risk factors, such as heavy lifting, bending, overhead reaching, pushing and pulling, repetitive tasks, or awkward body postures [19,20,21,22,23,24]. The susceptibility to MSS and MSDs can also be heightened by increased chronic stress in the workplace [25,26].

Health impairment also encompasses stress. Of the few studies conducted in the general mining industry, researchers found an association between long work hours and stress, e.g., mental fatigue and depressive symptoms, which, in turn, made workers more susceptible to injuries [11,27,28,29]. Long work hours and overtime also impact psychosocial factors of work, such as increased job dissatisfaction and decreased productivity [12,13,14,15,16,17]. According to the JD-R model, stress and psychosocial factors, such as job dissatisfaction and decreased productivity, can lead to burnout [8,16,30]. Burnout, as recently defined by the International Disease Classification (ICD-11), is a syndrome that results from unmanaged chronic workplace stress that can result in health impairment, such as energy depletion, exhaustion, mental distance, and reduced productivity [31]. Burnout can extend to all occupations—beyond healthcare workers, police officers, and firefighters—including SSGM workers.

As of 2019, the median age of U.S. nonmetallic mining and quarrying workers, i.e., SSGM was 48 years old [32,33]. In a meta-analysis analyzing employee burnout and its relationship between age or years of experience, the researchers found a small negative correlation between some U.S. employees’ age and emotional exhaustion—a component of burnout—as well as a small negative correlation between years of experience and emotional exhaustion [34]. Recent research also suggests age associations with burnout, especially when age begins to negatively impact workers’ physical and psychological abilities [35,36]. Work tasks at SSGM operations are demanding, particularly since operations are production-focused [37]. Workers in underground coal, surface coal, surface stone, and stone processing plants experience the greatest number of MSDs compared to other mining operations, potentially contributing to burnout experienced with age [22].

Evidence suggests long work hours and overtime can cause health impairment, stress, and burnout. However, to the best of our knowledge, this has not been assessed in SSGM workers despite their aforementioned elevated risk factors. A review on long work hours and wellbeing suggests that the effects of long work hours can be nuanced and vary significantly for different working populations [38]. Hence, we sought to explore the relationship between long work hours and overtime and health impairment in a sample of U.S. SSGM workers. This study is designed to test whether there is a difference in stress, burnout, and health status among SSGM workers who work 40 h or less each week, 41 to 50 h each week, 51 to 60 h each week and 60 h or more each week.

## 2. Materials and Methods

### 2.1. Participants

Survey data were collected from 459 full-time SSGM workers in the Midwest from fall 2019 to spring 2020, just prior to the World Health Organization declaring COVID-19 as a pandemic [39]. SSGM companies included were small to medium-sized businesses. Convenience sampling was used. Data were collected from participants completing their U.S. Mine Safety and Health Administration (MSHA) annual refresher training for their employer or at a training facility. All participants attending the training sessions visited were invited to participate. Participation in this cross-sectional study was voluntary. No personally identifiable information was collected. Consent was obtained prior to participation. More than 99% of the training attendees opted to participate. Workers were given a $20 gift card for participating. This study was deemed exempt by the Indiana University Institutional Review Board (Protocol # 1902635452).

### 2.2. Measures

Outcome variables in the study included stress, burnout, and a single-item measure of health status. This study adapted the Dutch Musculoskeletal Questionnaire (DMQ), which has been used extensively in similar research [40]. Further details on DMQ measures and SSGM outcomes are detailed in our work elsewhere [20,21]. The DMQ, aside from musculoskeletal symptom questions, includes a single-item health question that was adopted. Additional health measures were not added as single-measure health items have been found valid [41,42,43]. The single-item measure asked participants to respond to the following item: “In general, would you say your health is [poor/fair/good/very good/excellent]”.

Stress and burnout measures were derived from existing valid and reliable scales used in workplace research. Stress was comprised of six items derived from the work of DeJoy and colleagues [44]. Sample items included, “in the last month, how often have you felt nervous and stressed because of work” and “in the last month, how often have you felt you were unable to control the important things at work.” Items were assessed using a 5-point Likert-type scale with response options from *almost never* to *almost always* (Cronbach’s α = 0.89).

Burnout was assessed using a 10-item scale from Malach-Pines [45]. The stem stated, “When you think about your work overall, how often do you feel the following?” with participants responding to items such as tired, helpless, hopeless, and trapped. Items were assessed on a 5-point Likert-type scale with response options from *almost never* to *almost always* (Cronbach’s α = 0.90).

For the study, the main independent variable of interest was work hours. Participants were asked to report how many hours they worked each week, including overtime. Hours were transformed for this analysis into four categories: Up to and including 40 h, 41 to 50 h, 51 to 60 h, and greater than 60 h. Work hours were included as a categorical variable in each of the ANOVA models. Participants’ responses were categorized into male and female. Responses for age were categorized into less than 25, 25 to 34, 35 to 44, 45 to 54, 55 to 64, and 65 or older. Body mass index (BMI) was categorized using standard classifications, including underweight, normal weight, overweight, and obese.

### 2.3. Data Analysis

Three univariate analyses (ANOVA’s) were completed for the purposes of this study. Outcome variables included stress, burnout, and health status. Each analysis included the categorical variables age, sex, and work hours. The independent variable of interest was work hours. Age and sex were included as control variables. Additionally, the categorical variable BMI was added to the health status analysis as a control variable. Additionally, interaction terms were included in each of the analyses. Results for each of the ANOVA analyses are detailed in Table 1. Bonferroni post hoc tests were completed to further assess significant relationships identified in the analysis of variance. Probability values of ≤0.05 were considered statistically significant. All analyses were performed using SPSS v28 (IBM Corp., Armonk, NY, USA).

## 3. Results

The mean age was 45 (SD = 13.91). Ninety-three percent of the participants reported their sex as male (*n* = 423), and seven percent reported their sex as female (*n* = 31). BMI was calculated from participant self-reported height and weight. The mean BMI was 29.77 (SD = 6.1), which falls into the overweight range.

For stress, we found a statistically significant difference in stress according to work hours *F*(3, 404) = 3.848, *p* = 0.010. Bonferonni post hoc tests showed a significant difference between those working 60 h or more and those working between 51 and 60 h, with a 0.435 average difference of stress perceptions (*p* = 0.016), a significant difference between those working 60 h or more and those working between 41 to 50 h, with a 0.534 average difference of stress perceptions (*p* < 0.001), and a significant difference between those working 60 h or more and those working 40 h or less, with a 0.777 average difference of stress perceptions (*p* < 0.001).

For burnout, we found a statistically significant difference in burnout according to work hours *F*(3, 409) = 6.521, *p* < 0.001. Bonferonni post hoc tests showed a significant difference between those working 60 h or more and those working between 51 and 60 h, with a 0.501 average difference of burnout perceptions (*p* < 0.001), a significant difference between those working 60 h or more and those working between 41 to 50 h, with a 0.504 average difference of burnout perceptions (*p* < 0.001), and a significant difference between those working 60 h or more and those working 40 h or less, with a 0.706 average difference of burnout perceptions (*p* < 0.001).

For health status, we found a statistically significant difference in health status according to work hours *F*(3, 368) = 5.007, *p* = 0.002. Bonferonni post hoc tests showed a significant difference between those working 60 h or more and those working 40 h or less. The mean health status for those working 60 h or more was 0.376 less than those working 40 h or less (*p* = 0.049). In addition to work hours, it was determined the interaction between hours worked, and BMI was significant *F*(3, 368) = 3.146, *p* = 0.005, inferring the impact of BMI depends on the level of hours worked (Table 1). Neither age nor sex was significant in the analyses.

## 4. Discussion

Long work hours and overtime have been linked to health impairment among workers, which encompasses both physical, e.g., MSS and MSDs, and mental health impairment, e.g., stress. Among Spanish white-collar workers, who worked 51–60 h a week, noticeable poor health outcomes were reported such as poor mental health, hypertension, lack of sleep, and limited time for physical activity [13]. In the vein of overall health impairment, Dembe and colleagues found jobs with overtime schedules resulted in 61% higher injury rates compared to those without overtime; working at least 12 h a day was associated with a 37% increased hazard rate and working more than 60 h per week was associated with a 23% increased hazard waste in a U.S. nationally representative sample [15]. In other fields that experience long work hours and overtime, such as emergency medical services and clinicians, longer work hours resulted in stress, fatigue, and burnout [46,47]. Furthermore, working more than 12 h per shift was found to have an exponentially negative effect on productivity and burnout, particularly if employees perceived low autonomy [48]. A study on Finnish hospital employees who worked more than 48 h per week shows that they experienced job strain, sleep complaints, depressive symptomology, and increased sick absences [49]. Clearly, long work hours and overtime are detrimental to overall health and contribute to stress and burnout in several working populations, including SSGM workers. Our findings among this U.S. SSGM sample of workers are concordant with this literature and provide additional evidence of the harmful effects of regularly working long hours.

The results of the present study suggest that working more than 60 h each week was most problematic, accounting for significant differences between groups. Post hoc analyses illustrated that overall health was significantly worse for those working more than 60 h each week compared to those working 40 h or less each week. More alarming was that post hoc analyses determined both stress and burnout were significantly greater for workers working more than 60 h each week compared to all other groups.

Age and sex were included in each of the analyses as control variables. There were no significant differences between age groups for each of the outcomes. Similarly, no significant differences were found between males and females for each of the outcomes.

BMI was included in the health status analysis as a control variable. Although previous literature used BMI to define anthropometric height/weight characteristics in adults and use it as a proxy for an individual’s overweight/obesity level due to their height to weight ratio [50,51,52], a critical review of BMI and recent literature found BMI to be a poor indicator of body fat percentage, did not take into account other health issues, or consider social determinants of health that contribute to weight [51,52,53]. This may be the case in our study. The mean values for health status were not significantly different between BMI groups. However, there was a significant interaction relationship between the interaction of BMI and work hours. In the present study, this implies the BMI’s impact on health is dependent upon work hours. This relationship considers the main health issue in our research and again illustrates the relevance of work hours on health impairment.

While the obvious recommendation to reduce SSGM workers’ health impairment is to eliminate long work hours and overtime entirely, that may be an untenable option. Because of personal economic circumstances, workers may opt to work long hours and overtime despite the known detrimental long-term health effects. Thus, solutions to minimize long work hours and overtime must strike the balance of not only protecting workers but also seeking their input. Possible solutions include restricting frequent overtime or back-to-back shifts of long work hours (>10), so workers have sufficient time for external recovery (sleep). Employers should also consider facilitating sufficient internal recovery hours, e.g., longer breaks and rest periods [48]. Employers should also evaluate their management style to determine if workers can be given more latitude. Based on the JD-R model, workers with greater perceived autonomy may experience less stress and burnout due to the moderating effect of autonomy [8,48].

The implementation of ergonomic programs may be another way to address burnout. Ergonomic programs not only address physical injury exposures but other health impairment outcomes, such as burnout. According to the JD-R model, ergonomic programs would be considered a resource that buffers or controls demands, which helps minimize health impairment and bolsters motivation [54,55]. A systematic review of workplace interventions found that multi-component programs were more likely to result in improvements in mental and musculoskeletal system health among older employees. Furthermore, organizational-level programs and interventions were found to be most effective rather than placing the onus on individual-level or worker-level interventions [56]. Organizational-level solutions include increasing wages to reduce the need to work overtime, re-organizing work to avoid frequent overtime, e.g., staggering shifts, employing more staff to decrease job strain and the burden of long hours among workers, encouraging active participation in unions, providing health promotion and ergonomic programs, and, providing medical evaluations for employees [48,57].

Future research should build upon and extend the present study. Longitudinal assessments of the impact of long work hours and overtime on SSGM workers would provide more robust data and would allow causation to be inferred. Additionally, a more comprehensive assessment of individual employees and the work context that mitigates or exacerbates the impact of long work hours on health outcomes is warranted [38]. Additional data beyond BMI, e.g., lifestyle, diet, pre-existing conditions, and biomarkers, may provide further, specific evidence of factors that influence health impairment in the context of long work hours and overtime. Psychosocial factors, such as job satisfaction and safety climate, have been assessed among SSGM workers in another context [58]. These factors could be explored as possible job resources to determine if they buffer the effects of long work hours on stress, burnout, and health impairment outcomes [54,55]. Additionally, job control/autonomy could be assessed in the same context [55]. Lastly, more workplaces are moving toward a Total Worker Health^®^ (TWH) [59] approach. According to the National Institute for Occupational Safety and Health (NIOSH), TWH is defined as, “policies, programs, and practices that integrate protection from work-related safety and health hazards with the promotion of injury prevention efforts to advance worker well-being” [59]. TWH strategies integrate holistic health promotion education, resources, and tools for workers and management with more traditional health protection or safety strategies. Evidence of this organizational-level approach to promoting worker health, safety, and wellbeing is well-documented [60,61,62]. To our knowledge, no TWH approaches have been implemented among SSGM worksites, but research points to successful implementation in the construction industry to positively impact worker safety, personal health, and overall wellbeing [63,64,65].

### Limitations

This study is not without its limitations. First, this study used a cross-sectional design, so causality cannot be inferred. Second, participants were SSGM employees from the U.S. Midwest, therefore, the results may not be generalizable to miners outside of SSGM, in underground SSGM, or in other regions of the United States. Third, despite the survey being anonymous, there could be a potential for social desirability bias in the responses. Fourth, all health status was self-reported and not assessed prospectively after data collection but was collected at the time of the survey.

## 5. Conclusions

The present research described the impact of long work hours and overtime on the health impairment of a sample of U.S. SSGM workers. It is evident long work hours and overtime, particularly working more than 60 h each week, are associated with increased stress and burnout and decreased overall health of these workers. This novel research has implications for future research and practice for this mining sector. Future research should build upon and extend the present study. Along with protective safety controls, mine administrators, safety administrators, and other mine organization leaders should enact organizational-level programs and changes to bolster and improve the health of the workforce in this mining sector so that long work hours are not distressing, do not lead to burnout, and do not diminish overall health. Although it seems the easiest change is to reduce work hours, this change is not easily implemented given production needs and reduced hours would reduce worker earnings, potentially impacting a worker’s livelihood. Collaborative solutions between management and workers are needed. Possible organizational-level solutions to consider might include increasing wages, re-organizing work to avoid long work hours and frequent overtime, employing more staff to decrease job strain, encouraging active participation in unions, providing health promotion and ergonomic programs, and conducting medical evaluations for employees [66]. As more workplaces are moving toward a Total Worker Health^®^ (TWH) [59] approach to protect and promote worker health, TWH strategies might also be considered.

## Figures and Tables

**Table 1 ijerph-19-07740-t001:** ANOVA results.

	*F* Statistic	Df1	Df2	*p*-Value
**Health**				
**BMI**	2.558	3	368	0.055
**Age**	0.909	5	368	0.475
**Gender**	1.518	1	368	0.219
**Hours Worked**	5.007	3	368	0.002 **
**Hours worked * BMI**	3.146			0.005 **
**Stress**				
**Age**	0.328	5	404	0.896
**Gender**	0.775	1	404	0.379
**Hours Worked**	3.848	3	404	0.010 **
**Burnout**				
**Age**	0.967	5	409	0.438
**Gender**	0.299	1	409	0.585
**Hours Worked**	6.521	3	409	<0.001 ***

*p* ≤ 0.05 *, *p* ≤ 0.01 **, *p* < 0.001 ***.

## Data Availability

The data presented in this study are available on request from the senior author. The data are not publicly available due to privacy.

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
