# Peer review of "Long Work Hours, Overtime, and Worker Health Impairment: A Cross-Sectional Study among Stone, Sand, and Gravel Mine Workers"

_ijerph, 2022, doi:10.3390/ijerph19137740_

Round 1
Reviewer 1 Report
1. The focus on stone, sand, and gravel mine (SSGM) workers is a strength of the study to focus on workers at high risk for occupational health concerns.
2.In general I found this paper to be conceptually and methodologically rather weak as a cross-sectional study. What is the novel contribution to the literature? How does this study contribute over and above existing research about work hours and health other than using this unique sample?
3. The introduction has a paragraph about MSDs. However, the connection between MSDs and the current study is not clear unless the link has to do with overall self-rated health.
3. The purpose of the study indicates the following purpose: "We sought to explore the relationship between long work hours and overtime, mental health, and health impairment in a sample of US SSGM workers." How does burnout relate to mental health? I think it would be more accurate to state burnout here in lieu of mental health.
4. In the results section I would like to see a table of descriptive statistics (mean, standard deviation) for all variables, as well as a correlation matrix prior to presenting the regression results.
5. What is the rationale for conducting the chi-square analyses, which is a non-parametric test, compared to an ordinal regression analysis?
6. The discussion suggests that ergonomic programs could help address burnout - how? More specificity about the type of program and explanation regarding the link between ergonomics program and burnout is necessary to make this portion of the paper clearer.
7. Minor comment: The following paper seems highly relevant to the current study. Perhaps this paper or some of those cited within this paper may be helpful to the authors:
Ganster, Daniel C., Christopher C. Rosen, and Gwenith G. Fisher. "Long working hours and well-being: What we know, what we do not know, and what we need to know." J Bus & Psych 33.1 (2018): 25-39.
8. The last comment about Total Worker Health is a good point, but very vague. I recommend flushing this out a bit more.
Reviewer 2 Report
I read the article entitled "Long work hours, overtime, and worker health impairment: A study among stone, sand and gravel mine workers". It's definitely an original study as I have not read other studies investigating burnout in this category of workers. The sad thing about this study is that participants were not given an additional general health questionnaire and the researchers limited themselves to looking at the issue, asking only one graded question. My observations follow:
- Introduction:
I would ask the authors to expand the Introduction. In particular, I would suggest adding information about burnout. The ICD-11 allows for burnout in workers of all occupations however the average reader perceives burnout in occupations such as health care workers, police officers, firefighters, etc.
- Materials and Methods
2.1. Participants
Please provide more information on how the sample of participants was selected. All participants who attended the retraining program were invited, some random selection was made by them, what percentage accepted the invitation, what was included in the compensation for their time?
2.2. Measures
I would ask the authors to add References that justify their choice to use a «single-item measure of health status».
Results
The results lack an important table that presents the descriptive characteristics of the participants (N, average, SD) please add that table.
Reviewer 3 Report
This article is so poorly written and confusing that it was difficult to evaluate. I have included some indication of where it could be fixed in the attached PDF, but a thorough re-write and edit is required. Not only are there extensive grammatical errors, but very often the wrong words and phrases are used. You really need to have a lot more people read it and provide feedback before submitting it to the journal. Additionally, I was fairly astonished that your conclusion is basically that workers need to toughen up or employers should only hire workers who can hack it as the solutions.

Round 2
Reviewer 1 Report
Thank you very much for your outstanding revisions to the paper. I appreciate your responsiveness to the previous review and recommendations.
Reviewer 3 Report
Writing problems and lack of clarity persist in this manuscript. There are several examples of non-conformity with minimal standards for academic writing (e.g., the future tense used in the abstract, the discussion not leading with findings, odd phrasing and word choice.) The underlying data and analysis are useful, however. I recommend that you circulate this manuscript for critique among your colleagues, revise it, and submitted to a different journal.
